# The Difference in Cytotoxic Activity between Two Optical Isomers of Gelsemine from *Gelsemium elegans* Benth. on PC12 Cells

**DOI:** 10.3390/molecules24102004

**Published:** 2019-05-25

**Authors:** Li Lin, Yan-Chun Liu, Zhao-Ying Liu

**Affiliations:** 1College of Veterinary Medicine, Hunan Agricultural University, Changsha 410128, China; linli.1110@foxmail.com (L.L.); lyc20180101@sina.com (Y.-C.L.); 2Hunan Engineering Technology Research Center of Veterinary Drugs, Hunan Agricultural University, Changsha 410128, China

**Keywords:** *Gelsemium elegans*, gelsemine, koumine, PC12 cells

## Abstract

Two optical isomers, +/− gelsemine (1, 2), together with one known compound were isolated from the whole plant of *G. elegans*. The structures of the separated constituents were elucidated on 1D and 2D (^1^H-^1^H COSY, HMBC, HSQC) NMR spectroscopy and high-resolution mass spectrometry (HRMS). The isolated alkaloids were tested in vitro for cytotoxic potential against PC12 cells by the MTT assay. As a result, (+) gelsemine (compound **1**) exhibited cytotoxic activity against PC12 cells with an IC_50_ value of 31.59 μM, while (−) gelsemine (compound **2**) was not cytotoxic.

## 1. Introduction

*Gelsemium elegans* Benth. (*G. elegans*), referred to as “Gou-Wen” and “Duan-Chang-Cao” in China, belongs to *Loganiaceae* family and is a famous toxic plant that is widely distributed in Southeast Asia and southern China [1]. This plant has been used as a traditional medicine for the treatment of pain, spasticity, skin ulcers, migraines, neuralgia, sciatica, cancer, and various types of sores [2,3,4,5,6,7,8]. The *Gelsemium* genus includes three species, all of which have a large number of alkaloids, including indole, bisindole, and monoterpenoid alkaloids [7,9,10,11]. To date, more than 120 kinds of alkaloids, including some stereo-isomer compounds, have been isolated from *G. elegans*. Most of the alkaloids are classified into five types based on their chemical structure characteristics: gelsedine-, koumine-, humantenine-, sarpagine-, and gelsemine-type [5,12]. The *Gelsemium* alkaloids have potent cytotoxic, analgestic, anti-inflammatory, immunomodulating, and antiarrhythmic activities, which means they have potential as new drugs [7,13]. Their tremendous effects and the unusual and densely functionalized nature of the hexacyclic structure have been used in the development of synthetic approaches toward gelsemine, humantenine, and gelsedine-type alkaloids. The first total syntheses of (±)-gelsemine were disclosed in 1994 by the groups of Johnson. Gelsemine and koumine of *G. elegans* inhibited *Tetrahymena thermophila* cells’ growth in a dose-dependent manner [14,15]. In the present study, two optical isomers, +/− gelsemine (1, 2), together with one known compound, koumine, were isolated from the whole plant of *G. elegans* (Figure 1). According to the structure of gelsemine, gelsemine has seven chiral carbon and two chiral nitrogen atoms. The structures of these compounds were elucidated mainly by NMR spectroscopic and high resolution mass spectroscopic methods. Furthermore, all compounds have been tested in vitro for cytotoxic potential against PC12 cells by MTT assay.

## 2. Results and Discussion

Compound **1** was isolated as a white powder (for ^1^H and ^13^C-NMR, see Table 1). The molecular formula was determined as C_20_H_23_N_2_O_2_ by HR-ESI-MS in positive mode (*m/z* 323.1758 [M + H]^+^, calcd for [M + H]^+^ 323.1751). The ^1^H and ^13^C-NMR (See Appendix A) profile of compound **1** matched with the reported data for gelsemine [16].

Compound **2** was isolated as a yellow powder and had similar NMR data to **1** (Table 1), but the purity of compound **2** was lower than **1**. The molecular formula was determined as C_20_H_23_N_2_O_2_ by HR-ESI-MS in positive mode (*m*/*z* 323.1763 [M + H]^+^, calcd for [M + H]^+^ 323.1751). The retention time of compound **1** and **2** was 16.861 min and 16.619 min, respectively. The MS^2^ spectra of compounds **1** and **2** are shown in Figure 2. The product ions of compounds **1** and **2** were quite similar. According to the data of ^1^H and ^13^C-NMR(See Appendix A), compound **2** was identified as gelsemine. 

Compound **3** was isolated as a yellow powder with the following properties: positive ESI-MS *m*/*z*: 307.1812 [M + H]^+^, ^13^C-NMR (See Appendix A) (101 MHz, MeOD) δ 187.85 (2-C), 155.04 (13-C), 144.59 (8-C), 137.95 (19-C), 129.51 (11-C), 127.72 (10-C), 124.58 (9-C), 121.47 (12-C), 116.93 (18-C), 71.88 (3-C), 61.91 (17-C), 58.97 (7-C), 58.15 (21-C), 57.77 (5-C), 46.59 (20-C), 42.84 (N-Me), 38.52 (16-C), 33.97 (15-C), 30.46 (6-C), 25.93 (14-C). ^1^H-NMR(See Appendix A) (400 MHz, MeOD) δ 7.65 (1H, d, *J* = 7.3 Hz, 12-H), 7.55 (1H, d, J = 7.6 Hz, 9-H), 7.40 (^1^H, td, *J* = 7.6, 1.2 Hz, 11-H), 7.33 (1H, td, *J* = 7.5, 1.0 Hz, 10-H), 4.92 (1H, dd, J = 8.3, 1.7 Hz, 18α-H), 4.82 (dd, *J* = 11.2, 0.8 Hz, 1H, 18β-H), 4.69 (1H, dd, *J* = 17.6, 11.3 Hz, 19-H), 4.32 (1H, dd, *J* = 12.0, 4.6 Hz, 17α-H), 3.64 (1H, d, J = 12.0 Hz, 17β-H), 3.29 (1H, d, *J* = 11.9 Hz, 21α-H), 3.03 (1H, m, 21β-H), 2.88 (1H, m, 5-H), 2.67 (1H, s, *N*-Me), 2.42 (2H, m, 6-H), 1.87 (dt, *J* = 14.8, 2.0 Hz, 1H, 14β-H). This profile was consistent with the reported NMR data for koumine [16]. 

The effects of the three compounds on the activity of poorly differentiated PC-12 cells and highly differentiated PC-12 cells (Figure 3) were investigated, and it was found that only compound **1** inhibited viability in highly differentiated PC-12 cells, while the other compounds were not toxic to either type of PC-12 cells. As shown in Table 2, compound **1** on poorly differentiated PC-12 cells showed no toxicity (IC_50_ > 100 μM), and did not affect its activity. However, compound **1** showed toxicity to highly differentiated PC-12 cells and decreased cell activity, and its IC_50_ was 31.59 μM. The results indicated that compound 1 exhibited some cytotoxic activity against PC12 cells, while compound 2 did not. 2D (^1^H-^1^H COSY, HMBC, HSQC) NMR spectroscopy (Appendix A) showed that compounds **1** and **2** are quite similar. However, the optical rotation results showed that compound **1** and **2** are two optical isomers of gelsemine (**1**: _[α]25_D = 793.79 (c = 0.047, MeOH), **2**:_[α]25_D = −909.09 (c = 0.025, MeOH)) which explained why they differ in cytotoxic activity on PC12 cells.

## 3. Materials and Methods

### 3.1. General Experimental Procedures

NMR spectra were acquired using a Bruker ACF-400 spectrometer (the ^1^H-NMR spectra at 400 MHz and ^13^C-NMR spectra at 100 MHz, (Bruker, Rheinstetten, Germany). The mass spectra were determined by an Agilent 1290 HPLC system (Agilent Technologies, California, CA, USA) coupled with a 6530 Q-TOF/MS accurate-mass spectrometry. Silica gel (100–200, 200–300, 300–400 mesh) was used for column chromatography (CC) and silica GF254 for TLC was supplied by Qingdao Haiyang Chemical Factory, Qingdao, RP China. All solvents used were of analytical grade and obtained from Shanghai Chemical Reagents Company, Shanghai, RP China. Acetonitrile and methanol used for chromatographic grade were purchased from Merck, German. PC12 cells were purchased from Cell bank of Chinese academy of sciences. Staurosporine (STSP) was obtained from MedChem Express, Shanghai, China.

### 3.2. Plant Material

The whole plant of *G. elegans* was collected from Longyan city, Fujian province (24.699739 N, 116.98967 E). The samples were authenticated by Dr Qi Tang at Hunan Agricultural University.

### 3.3. Extraction and Isolation

The air-dried whole plant of *G. elegans* (100 kg) was powdered and extracted two times (2 × 24 h) with 0.5 % sulfuric acid solution (100 L) at room temperature. The extracted solution was filtered through a 200-mesh filter and combined, then was basified with NaOH (8 mol/L) to pH 7.0, and dried under reduced pressure. Then, the fraction (1 kg) was extracted by reflux in 1.5 L of 95 % ethanol for twice times (each for 2 h). After filtration, the extract was combined and concentrated under reduced pressure. The ethanol fraction (138 g) was further fractionated through a silica gel column chromatography (CC), eluting with CH_2_Cl_2_–MeOH (from 100:0 to 0:100, v/v) to obtain 14 fractions according to TLC analysis (Frs. 1–14). Fraction 7 (2.85 g) was subjected to silica gel CC (CH_2_Cl_2_-MeOH -ammonia, 95:5:0.05, v/v/v) to afford compound **1** (78 mg). Fraction 3 (5.00 g) was subjected to silica gel CC (CH_2_Cl_2_-MeOH-ammonia, 95:5:0.05–90:10:0.05, v/v/v) to afford three subfractions (Fr. 3-1 and Fr. 3-3). Fr. 3-1 (0.50 g) was subjected to silica gel CC (CH_2_Cl_2_–MeOH-ammonia, 20:0.8:0.05, v/v/v) to obtain compound **2** (71 mg) and compound **3** (99 mg). 

### 3.4. Cell Line and Culture

Poorly differentiated PC-12 cells: PC12 cells (Cell bank of Chinese academy of sciences) were grown in 100 cm^2^ culture flasks (1 × 10^6^ cells per flask) at 37 °C in serum-containing medium, 85% RPMI 1640, 10% heat-inactivated horse serum (Gibco, Grand Island, NY, USA ), 5% fetal calf serum (Biowest, Nuaillé, France ), penicillin & streptomycin (1× P/S, Procell, Tongxiang, China, Lot: 16050130) and no NGF. Culture medium was exchanged three times weekly. Highly differentiated PC-12 cells: PC-12 cells were grown in serum-containing medium (90% RPMI 1640, 10% fetal calf serum, 1× P/S), treated with 20 ng/mL of NGF (Sigma, San Francisco, CA, USA, Lot: SRP3018). The medium was changed every 2–3 days and NGF was added if required.

## Figures and Tables

**Figure 1 molecules-24-02004-f001:**
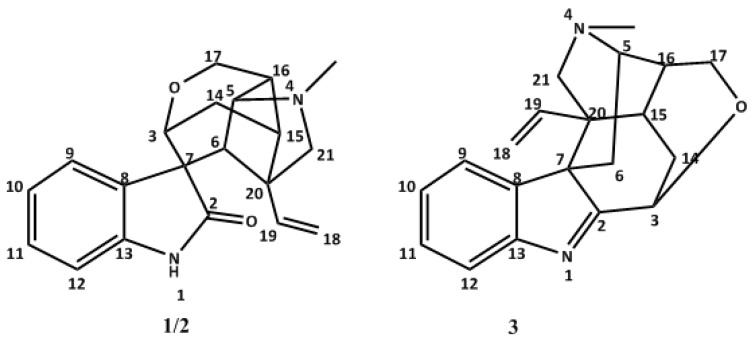
The structures of compound (±)-gelsemine (**1**/**2**) and koumine (**3**).

**Figure 2 molecules-24-02004-f002:**
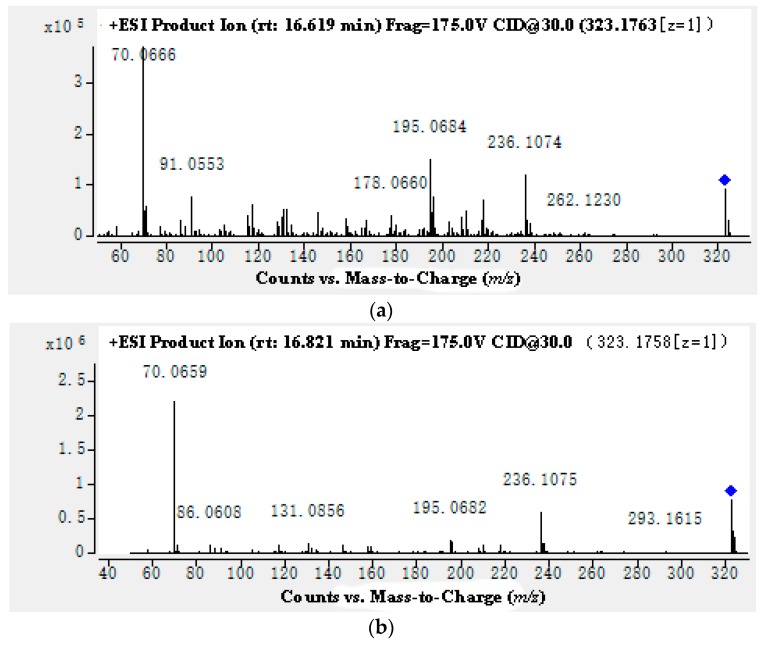
The MS/MS spectra of compound **1** (**a**) and **2** (**b**).

**Figure 3 molecules-24-02004-f003:**
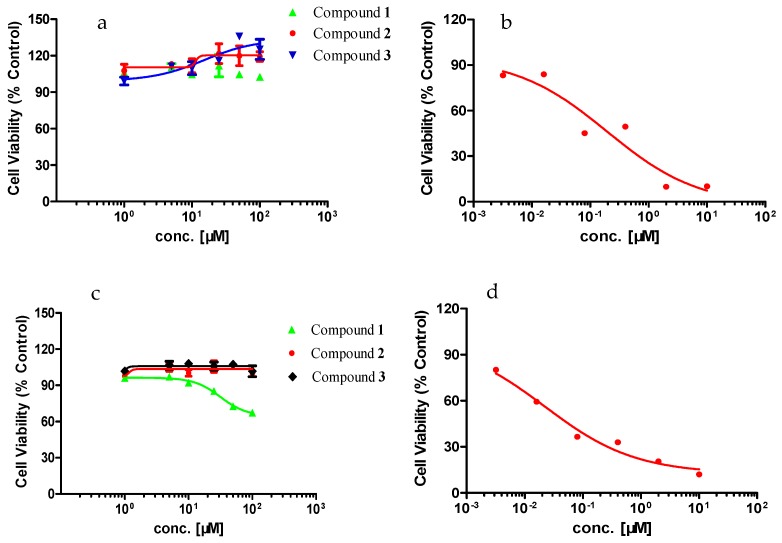
The cell viability of the three compounds in a series of concentrations (5, 10, 25, 50, 100 μM) and positive drug staurosporine (STSP) on two types of PC-12 cells after exposure for 48 h. (**a**) the three compounds’ concentration responses of poorly differentiated PC-12 cells; (**b**) STSP concentration response in poorly differentiated PC-12 cells; (**c**) the three compounds’ concentration responses in highly differentiated PC-12 cells; (**d**) STSP concentration response in highly differentiated PC-12 cells.

**Table 1 molecules-24-02004-t001:** ^1^H-NMR (400 MHz) and ^13^C-NMR (100 MHz) data for Compound **1** and **2** in MeOD (δ in ppm and *J* in Hz).

Position	Compound 1		Compound 2	
	δ_C_	δ_H_	δ_C_	δ_H_
2	179.26	/	180.56	/
3	69.58	3.60 (1H, s)	70.97	3.64 (1H, s)
5	71.81	4.12 (1H, d, 11.0)	73.28	4.13 (1H, d, 11.0)
6	50.63	1.99 (1H, s)	51.93	2.01 (1H, s)
7	54.19	/	55.55	/
8	131.70	/	133.00	/
9	127.99	7.46 (1H, d, 7.6)	129.30	7.46 (1H, d, 7.6)
10	121.23	7.00 (1H, t, 7.6)	122.65	7.00 (1H, t, 7.6)
11	128.10	7.22 (1H, t, 7.7)	129.49	7.22 (1H, t, 7.7)
12	108.97	6.87 (1H, d, 7.7)	110.40	6.87 (1H, d, 7.7)
13	141.20	/	142.60	/
14	22.34	2.04 (1H, m)	23.70	2.04 (1H, m)
		2.84 (1H, dd, 14.3, 2.1)		2.84 (1H, m)
15	35.63	2.34 (1H, t)	36.98	2.38 (1H, s)
16	37.50	2.49 (1H, m)	38.97	2.50 (1H, m)
17	60.81	3.97 (1H, d, 11.0)	62.14	3.97 (1H, m)
		3.72 (1H, s)		3.72 (1H, s)
18	111.59	4.98 (1H, d, 17.8)	113.16	4.98 (1H, d, 17.8)
		5.05 (1H, d, 11.0)		5.06 (1H, d, 11.0)
19	138.31	6.26 (1H, dd, 17.8,11.0)	139.48	6.25 (1H, dd,17.8, 11.0)
20	53.68	/	55.02	/
21	65.17	2.78 (1H, d,10.8)	66.45	2.80 (1H, d, 10.7)
		3.33 (1H, d, 1.3)		/
*N*-CH3	39.03	2.28 (3H,s)	40.43	2.31 (3H, s)

**Table 2 molecules-24-02004-t002:** Cytotoxicity of compounds **1**–**3** against PC12 cell lines (IC_50_, μM)

Compound	PC12 Highly Differentiated Cells	PC12 Poorly Differentiated Cells
**1**	>100	31.59
**2**	>100	>100
**3**	>100	>100
STSP	0.1944	0.022

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
