# Peer review of "The Difference in Cytotoxic Activity between Two Optical Isomers of Gelsemine from Gelsemium elegans Benth. on PC12 Cells"

_molecules, 2019, doi:10.3390/molecules24102004_

Round 1

Reviewer 1 Report

Your english could be improved, also dont forget to mentio most recent publicashions, such as:

Effects of gelsemine on oxidative stress and DNA damage responses of Tetrahymena thermophila.

quality of the figures (1 and 3) could be improved

Author Response

1. Your English could be improved, also don’t forget to mention most recent publications, such as: effects of gelsemine on oxidative stress and DNA damage responses of Tetrahymena thermophila.

Answer: Thank you very much for reviewing our manuscript and giving us valuable suggestions. Our manuscript was revised the language by the American Journal Experts. We also provide the editorial certificate for you in the online submission system. Also, we added the reference in the revised manuscript (see ref. 14-15).

2. Quality of the figures (1 and 3) could be improved

Answer: Thank you very much for suggestion. We have re-processed the figures 1 and 3.

Reviewer 2 Report

Abstract section:

Please report "in vitro" and "G. elegans" in italics.

Introduction section:

Please correct the typo ralted to the name of the alkaloid at line 36.

Throughout manuscript:

Please make uniform the name of the plant. I suggest to use the name, in extenso and in italics, at the first appearance in the text. Then the authors should always refer to plant with the name "G. elegans", always in italics. When they use expression like Gelsemium  genus and Gelsemium alkaloids, the use of italics is strongly suggested.

Figure 2:

Chinese characters should be substituted by latin characters.

Legend of Figure 3:

Please edit the legend. The sentences need revision. The authors should clearly describe the effects of each compounds on PC12 viability.

Line2 118-119: Correct the sentence in analogy with that reported in line 114-115.

As regards in vitro stuties, please substitue "dose-response" with "concentration-response".

Paragraph 3.2:

Please include the GPS coordinates related to the place of collection. Additionally, it is reccomended to include the name of the botanist that authenticated the plant material.

Author Response

Review 2 comments:

1. Abstract section: Please report "in vitro" and "G. elegans" in italics.

Answer: Thank you very much for good suggestion. We have corrected the format according to your comment in the revised manuscript.

2. Introduction section: Please correct the typo related to the name of the alkaloid at line 36.

Answer: Thank you very much for suggestion. We have corrected the name according to your comment in the revised manuscript (line 38).

3. Throughout manuscript: Please make uniform the name of the plant. I suggest to use the name, in extenso and in italics, at the first appearance in the text. Then the authors should always refer to plant with the name "G. elegans", always in italics. When they use expression like Gelsemium genus and Gelsemium alkaloids, the use of italics is strongly suggested.

Answer: Thank you very much for good suggestion. We have corrected the expression according to your comment in the revised manuscript with yellow highlight.

4. Figure 2: Chinese characters should be substituted by latin characters.

Answer: Thank you very much for suggestion. We have re-processed th e Figure 2.

5. Legend of Figure 3: Please edit the legend. The sentences need revision. The authors should clearly describe the effects of each compounds on PC12 viability.

Answer: Thank you very much for suggestion. We have revised the legend of Figure 3 in the revised manuscript. (lines 86-90)

6. Line2 118-119: Correct the sentence in analogy with that reported in line 114-115.

Answer: Thank you very much for suggestion.We checked the original record and found the culture condition of PC-12 poorly cells and PC-12 highly cells were correct.

7. As regards in vitro stuties, please substitue "dose-response" with "concentration-response".

Answer: Thank you very much for suggestion. We have revised the expression according your comments in the revised manuscript. (lines 86-90)

8. Paragraph 3.2: Please include the GPS coordinates related to the place of collection. Additionally, it is reccomended to include the name of the botanist that authenticated the plant material.

Answer: Thank you very much for suggestion. We have added the GPS coordinates related to the place of collection and the name of the botanist that authenticated the plant material at lines 105-106.

Round 2

Reviewer 2 Report

Manuscript quality has been improved.